# Better Approximation and Faster Algorithm Using the Proximal Average

**Yaoliang Yu**
Department of Computing Science, University of Alberta, Edmonton AB T6G 2E8, Canada
`yaoliang@cs.ualberta.ca`

## Abstract

It is a common practice to approximate "complicated" functions with more friendly ones. In large-scale machine learning applications, nonsmooth losses/regularizers that entail great computational challenges are usually approximated by *smooth* functions. We re-examine this powerful methodology and point out a *nonsmooth* approximation which simply pretends the linearity of the proximal map. The new approximation is justified using a recent convex analysis tool—proximal average, and yields a novel proximal gradient algorithm that is *strictly* better than the one based on smoothing, without incurring any extra overhead. Numerical experiments conducted on two important applications, overlapping group lasso and graph-guided fused lasso, corroborate the theoretical claims.

## 1 Introduction

In many scientific areas, an important methodology that has withstood the test of time is the approximation of "complicated" functions by those that are easier to handle. For instance, Taylor's expansion in calculus [1], essentially a polynomial approximation of differentiable functions, has fundamentally changed analysis, and mathematics more broadly. Approximations are also ubiquitous in optimization algorithms, *e.g.* various gradient-type algorithms approximate the objective function with a quadratic upper bound. In some (if not all) cases, there are multiple ways to make the approximation, and one usually has this freedom of choice. It is perhaps not hard to convince oneself that there is no approximation that would work best in all scenarios. And one would probably also agree that a specific form of approximation should be favored if it well suits our *ultimate* goal. Despite of all these common-sense, in optimization algorithms, the *smooth* approximations are still dominating, bypassing some recent advances on optimizing nonsmooth functions [2, 3]. Part of the reason, we believe, is the lack of new technical tools.

We consider the composite minimization problem where the objective consists of a smooth loss function and a sum of *nonsmooth* functions. Such problems have received increasing attention due to the arise of *structured sparsity* [4], notably the overlapping group lasso [5], the graph-guided fused lasso [6] and some others. These structured regularizers, although greatly enhance our modeling capability, introduce significant new computational challenges as well. Popular gradient-type algorithms dealing with such composite problems include the generic subgradient method [7], (accelerated) proximal gradient (APG) [2, 3], and the smoothed accelerated proximal gradient (S-APG) [8]. The subgradient method is applicable to any nonsmooth function, although the convergence rate is rather slow. APG, being a recent advance, can handle *simple* functions [9] but for more complicated structured regularizers, an inner iterative procedure is needed, resulting in an overall convergence rate that could be as slow as the subgradient method [10]. Lastly, S-APG simply runs APG on a smooth approximation of the original objective, resulting in a much improved convergence rate.

Our work is inspired by the recent advance on nonsmooth optimization [2, 3], of which the building block is the proximal map of the nonsmooth function. This proximal map is available in closed-form

for simple functions but can be quite expensive for more complicated functions such as a *sum* of nonsmooth functions we consider here. A key observation we make is that oftentimes the proximal map for each individual summand can be easily computed, therefore a bold idea is to simply use the sum of proximal maps, pretending that the proximal map is a linear operator. Somewhat surprisingly, this naive choice, when combined with APG, results in a novel proximal algorithm that is *strictly* better than S-APG, while keeping per-step complexity unchanged. We justify our method via a new tool from convex analysis—the proximal average [11]. In essence, instead of smoothing the nonsmooth function, we use a nonsmooth approximation whose proximal map is cheap to evaluate, after all this is all we need to run APG.

We formally state our problem in Section 2, along with the proposed algorithm. After recalling the relevant tools from convex analysis in Section 3 we provide the theoretical justification of our method in Section 4. Related works are discussed in Section 5. We test the proposed algorithm in Section 6 and conclude in Section 7.

## 2   Problem Formulation

We are interested in solving the following composite minimization problem:

$$\min_{x \in \mathbb{R}^d} \ell(x) + \bar{f}(x), \quad \text{where} \quad \bar{f}(x) = \sum_{k=1}^{K} \alpha_k f_k(x). \tag{1}$$

Here $\ell$ is convex with $L_0$-Lipschitz continuous gradient *w.r.t.* the Euclidean norm $\|\cdot\|$, and $\alpha_k \geq 0, \sum_k \alpha_k = 1$. The usual regularization constant that balances the two terms in (1) is absorbed into the loss $\ell$. For the functions $f_k$, we assume

**Assumption 1.** *Each $f_k$ is convex and $M_k$-Lipschitz continuous* w.r.t. *the Euclidean norm $\|\cdot\|$.*

The abbreviation $\overline{M^2} = \sum_{k=1}^{K} \alpha_k M_k^2$ is adopted throughout.

We are interested in the general case where the functions $f_k$ need not be differentiable. As mentioned in the introduction, a generic scheme that solves (1) is the subgradient method [7], of which each step requires merely an arbitrary subgradient of the objective. With a suitable stepsize, the subgradient method converges[1] in at most $O(1/\epsilon^2)$ steps where $\epsilon > 0$ is the desired accuracy. Although being general, the subgradient method is exceedingly slow, making it unsuitable for many practical applications.

Another recent algorithm for solving (1) is the (accelerated) proximal gradient (APG) [2, 3], of which each iteration needs to compute the proximal map of the nonsmooth part $\bar{f}$ in (1):

$$\mathsf{P}_{\bar{f}}^{1/L_0}(x) = \operatorname*{argmin}_{y} \tfrac{L_0}{2} \|x - y\|^2 + \bar{f}(y).$$

(Recall that $L_0$ is the Lipschitz constant of the gradient of the smooth part $\ell$ in (1).) Provided that the proximal map can be computed in constant time, it can be shown that APG converges within $O(1/\sqrt{\epsilon})$ complexity, significantly better than the subgradient method. For some simple functions, the proximal map indeed is available in closed-form, see [9] for a nice survey. However, for more complicated functions such as the one we consider here, the proximal map itself is expensive to compute and an inner iterative subroutine is required. Somewhat disappointingly, recent analysis has shown that such a two-loop procedure can be as slow as the subgradient method [10].

Yet another approach, popularized by Nesterov [8], is to approximate each nonsmooth component $f_k$ with a smooth function and then run APG. By carefully balancing the approximation and the convergence requirement of APG, the smoothed accelerated proximal gradient (S-APG) proposed in [8] converges in at most $O(\sqrt{1/\epsilon^2 + 1/\epsilon})$ steps, again much better than the subgradient method. The main point of this paper is to further improve S-APG, in perhaps a surprisingly simple way.

The key assumption that we will exploit is the following:

**Assumption 2.** *Each proximal map $\mathsf{P}_{f_k}^{\mu}$ can be computed "easily" for any $\mu > 0$.*

| **Algorithm 1:** PA-APG. | **Algorithm 2:** PA-PG. |
|---|---|
| 1: Initialize $x_0 = y_1$, $\mu$, $\eta_1 = 1$.<br>2: **for** $t = 1, 2, \dots$ **do**<br>3:    $z_t = y_t - \mu \nabla \ell(y_t)$,<br>4:    $x_t = \sum_k \alpha_k \cdot \mathsf{P}^\mu_{f_k}(z_t)$,<br>5:    $\eta_{t+1} = \frac{1 + \sqrt{1 + 4\eta_t^2}}{2}$,<br>6:    $y_{t+1} = x_t + \frac{\eta_t - 1}{\eta_{t+1}}(x_t - x_{t-1})$.<br>7: **end for** | 1: Initialize $x_0$, $\mu$.<br>2: **for** $t = 1, 2, \dots$ **do**<br>3:    $z_t = x_{t-1} - \mu \nabla \ell(x_{t-1})$,<br>4:    $x_t = \sum_k \alpha_k \cdot \mathsf{P}^\mu_{f_k}(z_t)$.<br>5: **end for** |

We prefer to leave the exact meaning of "easily" unspecified, but roughly speaking, the proximal map should be no more expensive than computing the gradient of the smooth part $\ell$ so that it does not become the bottleneck. Both Assumption 1 and Assumption 2 are satisfied in many important applications (examples will follow). As it will also become clear later, these assumptions are exactly those needed by S-APG.

Unfortunately, in general, there is no known *efficient* way that reduces the proximal map of the average $\bar{f}$ to the proximal maps of its individual components $f_k$, therefore the fast scheme APG is not readily applicable. The main difficulty, of course, is due to the nonlinearity of the proximal map $\mathsf{P}^\mu_f$, when treated as an operator on the function $f$. Despite of this fact, we will "naively" pretend that the proximal map is linear and use

$$\mathsf{P}^\mu_{\bar{f}} \overset{?}{\approx} \sum_{k=1}^{K} \alpha_k \mathsf{P}^\mu_{f_k}. \tag{2}$$

Under this approximation, the fast scheme APG can be applied. We give one particular realization (PA-APG) in Algorithm 1 based on the FISTA in [2]. A simpler (though slower) version (PA-PG) based on ISTA [2] is also provided in Algorithm 2. Clearly both algorithms are easily parallelizable if $K$ is large. We remark that any other variation of APG, *e.g.* [8], is equally well applicable. Of course, when $K = 1$, our algorithm reduces to the corresponding APG scheme.

At this point, one might be suspicious about the usefulness of the "naive" approximation in (2). Before addressing this well-deserved question, let us first point out two important applications where Assumption 1 and Assumption 2 are naturally satisfied.

**Example 1** (Overlapping group lasso, [5]). *In this example, $f_k(x) = \|x_{g_k}\|$ where $g_k$ is a group (subset) of variables and $x_g$ denotes a copy of $x$ with all variables not contained in the group $g$ set to 0. This group regularizer has been proven quite useful in high-dimensional statistics with the capability of selecting meaningful groups of features [5]. In the general case where the groups could overlap as needed, $\mathsf{P}^\mu_{\bar{f}}$ cannot be computed easily.*

*Clearly each $f_k$ is convex and 1-Lipschitz continuous w.r.t. $\|\cdot\|$, i.e., $M_k = 1$ in Assumption 1. Moreover, the proximal map $\mathsf{P}^\mu_{f_k}$ is simply a re-scaling of the variables in group $g_k$, that is*

$$[\mathsf{P}^\mu_{f_k}(x)]_j = \begin{cases} x_j, & j \notin g_k \\ (1 - \mu/\|x_{g_k}\|)_+ x_j, & j \in g_k \end{cases}, \tag{3}$$

*where $(\lambda)_+ = \max\{\lambda, 0\}$. Therefore, both of our assumptions are met.*

**Example 2** (Graph-guided fused lasso, [6]). *This example is an enhanced version of the fused lasso [12], with some graph structure exploited to improve feature selection in biostatistic applications [6]. Specifically, given some graph whose nodes correspond to the feature variables, we let $f_{ij}(x) = |x_i - x_j|$ for every edge $(i, j) \in E$. For a general graph, the proximal map of the regularizer $\bar{f} = \sum_{(i,j) \in E} \alpha_{ij} f_{ij}$, with $\alpha_{ij} \geq 0, \sum_{(i,j) \in E} \alpha_{ij} = 1$, is not easily computable.*

*Similar as above, each $f_{ij}$ is 1-Lipschitz continuous w.r.t. the Euclidean norm. Moreover, the proximal map $\mathsf{P}^\mu_{f_{ij}}$ is easy to compute:*

$$[\mathsf{P}^\mu_{f_{ij}}(x)]_s = \begin{cases} x_s, & s \notin \{i, j\} \\ x_s - \text{sign}(x_i - x_j) \min\{\mu, |x_i - x_j|/2\}, & s \in \{i, j\} \end{cases}. \tag{4}$$

*Again, both our assumptions are satisfied.*

Note that in both examples we could have incorporated weights into the component functions $f_k$ or $f_{ij}$, which amounts to changing $\alpha_k$ or $\alpha_{ij}$ accordingly. We also remark that there are other applications that fall into our consideration, but for illustration purposes we shall contend ourselves with the above two examples. More conveniently, both examples have been tried with S-APG [13], thus constitute a natural benchmark for our new algorithm.

## 3 Technical Tools

To justify our new algorithm, we need a few technical tools from convex analysis [14]. Let our domain $\mathcal{H}$ be a real Hilbert space with the inner product $\langle \cdot, \cdot \rangle$ and the induced norm $\| \cdot \|$. Denote $\Gamma_0$ as the set of all lower semicontinuous proper convex functions $f : \mathcal{H} \to \mathbb{R} \cup \{\infty\}$. It is well-known that the Fenchel conjugation

$$f^*(y) = \sup_x \langle x, y \rangle - f(x)$$

is a bijection and involution on $\Gamma_0$ (*i.e.* $(f^*)^* = f$). For convenience, throughout we let $\mathsf{q} = \frac{1}{2}\| \cdot \|^2$ (q for "quadratic"). Note that $\mathsf{q}$ is the only function which coincides with its Fenchel conjugate. Another convention that we borrow from convex analysis is to write $(f\mu)(x) = \mu f(\mu^{-1}x)$ for $\mu > 0$. We easily verify $(\mu f)^* = f^*\mu$ and also $(f\mu)^* = \mu f^*$.

For any $f \in \Gamma_0$, we define its Moreau envelop (with parameter $\mu > 0$) [14, 15]

$$\mathsf{M}_f^\mu(x) = \min_y \frac{1}{2\mu}\|x - y\|^2 + f(y), \tag{5}$$

and correspondingly the proximal map

$$\mathsf{P}_f^\mu(x) = \operatorname*{argmin}_y \frac{1}{2\mu}\|x - y\|^2 + f(y). \tag{6}$$

Since $f$ is closed convex and $\| \cdot \|^2$ is strongly convex, the proximal map is well-defined and single-valued. As mentioned before, the proximal map is the key component of fast schemes such as APG.

We summarize some nice properties of the Moreau envelop and the proximal map as:

**Proposition 1.** *Let* $\mu, \lambda > 0$, $f \in \Gamma_0$, *and* $\mathsf{Id}$ *be the identity map, then*

  i). $\mathsf{M}_f^\mu \in \Gamma_0$ *and* $(\mathsf{M}_f^\mu)^* = f^* + \mu\mathsf{q}$;

  ii). $\mathsf{M}_f^\mu \leq f$, $\inf_x \mathsf{M}_f^\mu(x) = \inf_x f(x)$, *and* $\operatorname{argmin}_x \mathsf{M}_f^\mu(x) = \operatorname{argmin}_x f(x)$;

  iii). $\mathsf{M}_f^\mu$ *is differentiable with* $\nabla \mathsf{M}_f^\mu = \frac{1}{\mu}(\mathsf{Id} - \mathsf{P}_f^\mu)$;

  iv). $\mathsf{M}_{\lambda f}^\mu = \lambda \mathsf{M}_f^{\lambda\mu}$ *and* $\mathsf{P}_{\lambda f}^\mu = \mathsf{P}_f^{\lambda\mu} = (\mathsf{P}_{f\lambda^{-1}}^\mu)\lambda$;

  v). $\mathsf{M}_{\mathsf{M}_f^\mu}^\lambda = \mathsf{M}_f^{\lambda+\mu}$ *and* $\mathsf{P}_{\mathsf{M}_f^\mu}^\lambda = \frac{\mu}{\lambda+\mu}\mathsf{Id} + \frac{\lambda}{\lambda+\mu}\mathsf{P}_f^{\lambda+\mu}$;

  vi). $\mu\mathsf{M}_f^\mu + (\mathsf{M}_{f^*}^{1/\mu})\mu = \mathsf{q}$ *and* $\mathsf{P}_f^\mu + (\mathsf{P}_{f^*}^{1/\mu})\mu = \mathsf{Id}$.

i) is the well-known duality between infimal convolution and summation. ii), albeit being trivial, is the driving force behind the proximal point algorithm [16]. iii) justifies the "niceness" of the Moreau envelop and connects it with the proximal map. iv) and v) follow from simple algebra. And lastly vi), known as Moreau's identity [15], plays an important role in the early development of convex analysis. We remind that $(\mathsf{M}_f^\mu)^*$ in general is different from $\mathsf{M}_{f^*}^\mu$.

Fix $\mu > 0$. Let $\mathsf{SC}_\mu \subseteq \Gamma_0$ denote the class of $\mu$-strongly convex functions, that is, functions $f$ such that $f - \mu\mathsf{q}$ is convex. Similarly, let $\mathsf{SS}_\mu \subseteq \Gamma_0$ denote the class of finite-valued functions whose gradient is $\mu$-Lipschitz continuous (*w.r.t.* the norm $\| \cdot \|$). A well-known duality between strong convexity and smoothness is that for $f \in \Gamma_0$, we have $f \in \mathsf{SC}_\mu$ iff $f^* \in \mathsf{SS}_{1/\mu}$, *cf.* [17, Theorem 18.15]. Based on this duality, we have the next result which turns out to be critical. (Proof in Appendix A)

**Proposition 2.** *Fix* $\mu > 0$. *The Moreau envelop map* $\mathsf{M}^\mu : \Gamma_0 \to \mathsf{SS}_{1/\mu}$ *that sends* $f \in \Gamma_0$ *to* $\mathsf{M}_f^\mu$ *is bijective, increasing, and concave on any convex subset of* $\Gamma_0$ *(under the pointwise order).*

It is clear that $\mathsf{SS}_{1/\mu}$ is a *convex* subset of $\Gamma_0$, which motivates the definition of the proximal average—the key object to us. Fix constants $\alpha_k \geq 0$ with $\sum_{k=1}^{K} \alpha_k = 1$. Recall that $\bar{f} = \sum_k \alpha_k f_k$ with each $f_k \in \Gamma_0$, *i.e.* $\bar{f}$ is the convex combination of the component functions $\{f_k\}$ under the weight $\{\alpha_k\}$. Note that we always assume $\bar{f} \in \Gamma_0$ (the exception $\bar{f} \equiv \infty$ is clearly uninteresting).

**Definition 1** (Proximal Average, [11, 15]). *Denote* $\mathbf{f} = (f_1, \ldots, f_K)$ *and* $\mathbf{f}^* = (f_1^*, \ldots, f_K^*)$. *The proximal average* $\mathsf{A}_{\mathbf{f},\boldsymbol{\alpha}}^{\mu}$, *or simply* $\mathsf{A}^{\mu}$ *when the component functions and weights are clear from context, is the unique function* $h \in \Gamma_0$ *such that* $\mathsf{M}_h^{\mu} = \sum_{k=1}^{K} \alpha_k \mathsf{M}_{f_k}^{\mu}$.

Indeed, the existence of the proximal average follows from the surjectivity of $\mathsf{M}^{\mu}$ while the uniqueness follows from the injectivity of $\mathsf{M}^{\mu}$, both proven in Proposition 2. The main property of the proximal average, as seen from its definition, is that its Moreau envelop is the convex combination of the Moreau envelops of the component functions. By iii) of Proposition 1 we immediately obtain

$$\mathsf{P}_{\mathsf{A}^{\mu}}^{\mu} = \sum_{k=1}^{K} \alpha_k \mathsf{P}_{f_k}^{\mu}. \tag{7}$$

Recall that the right-hand side is exactly the approximation we employed in Section 2.

Interestingly, using the properties we summarized in Proposition 1, one can show that the Fenchel conjugate of the proximal average, denoted as $(\mathsf{A}^{\mu})^*$, enjoys a similar property [11]:

$$\left[\mathsf{M}_{(\mathsf{A}^{\mu})^*}^{1/\mu}\right]\mu = \mathsf{q} - \mu\mathsf{M}_{\mathsf{A}^{\mu}}^{\mu} = \mathsf{q} - \mu\sum_{k=1}^{K} \alpha_k \mathsf{M}_{f_k}^{\mu} = \sum_{k=1}^{K} \alpha_k(\mathsf{q} - \mu\mathsf{M}_{f_k}^{\mu})$$

$$= \sum_{k=1}^{K} \alpha_k[(\mathsf{M}_{f_k^*}^{1/\mu})\mu] = \left[\sum_{k=1}^{K} \alpha_k \mathsf{M}_{f_k^*}^{1/\mu}\right]\mu,$$

that is, $\mathsf{M}_{(\mathsf{A}_{\mathbf{f},\boldsymbol{\alpha}}^{\mu})^*}^{1/\mu} = \sum_{k=1}^{K} \alpha_k \mathsf{M}_{f_k^*}^{1/\mu} = \mathsf{M}_{\mathsf{A}_{\mathbf{f}^*,\boldsymbol{\alpha}}^{1/\mu}}^{1/\mu}$, therefore by the injective property established in Proposition 2:

$$(\mathsf{A}_{\mathbf{f},\boldsymbol{\alpha}}^{\mu})^* = \mathsf{A}_{\mathbf{f}^*,\boldsymbol{\alpha}}^{1/\mu}. \tag{8}$$

From its definition it is also possible to derive an explicit formula for the proximal average (although for our purpose only the existence is needed):

$$\mathsf{A}_{\mathbf{f},\boldsymbol{\alpha}}^{\mu} = \left(\left(\sum_{k=1}^{K} \alpha_k \mathsf{M}_{f_k}^{\mu}\right)^* - \mu\mathsf{q}\right)^* = \left(\sum_{k=1}^{K} \alpha_k \mathsf{M}_{f_k^*}^{1/\mu}\right)^* - \mathsf{q}\mu, \tag{9}$$

where the second equality is obtained by conjugating (8) and applying the first equality to the conjugate. By the concavity and monotonicity of $\mathsf{M}^{\mu}$, we have the inequality

$$\mathsf{M}_{\bar{f}}^{\mu} \geq \sum_{k=1}^{K} \alpha_k \mathsf{M}_{f_k}^{\mu} = \mathsf{M}_{\mathsf{A}^{\mu}}^{\mu} \iff \bar{f} \geq \mathsf{A}^{\mu}. \tag{10}$$

The above results (after Definition 1) are due to [11], although our treatment is slightly different.

It is well-known that as $\mu \to 0$, $\mathsf{M}_f^{\mu} \to f$ pointwise [14], which, under the Lipschitz assumption, can be strengthened to uniform convergence (Proof in Appendix B):

**Proposition 3.** *Under Assumption 1 we have* $0 \leq \bar{f} - \mathsf{M}_{\mathsf{A}^{\mu}}^{\mu} \leq \frac{\mu\overline{M^2}}{2}$.

For the proximal average, [11] showed that $\mathsf{A}^{\mu} \to \bar{f}$ pointwise, which again can be strengthened to uniform convergence (proof follows from (10) and Proposition 3 since $\mathsf{A}^{\mu} \geq \mathsf{M}_{\mathsf{A}^{\mu}}^{\mu}$):

**Proposition 4.** *Under Assumption 1 we have* $0 \leq \bar{f} - \mathsf{A}^{\mu} \leq \frac{\mu\overline{M^2}}{2}$.

As it turns out, S-APG approximates the nonsmooth function $\bar{f}$ with the smooth function $\mathsf{M}_{\mathsf{A}^{\mu}}^{\mu}$ while our algorithm operates on the *nonsmooth* approximation $\mathsf{A}^{\mu}$ (note that it can be shown that $\mathsf{A}^{\mu}$ is smooth iff some component $f_i$ is smooth). By (10) and ii) in Proposition 1 we have

$$\mathsf{M}_{\mathsf{A}^{\mu}}^{\mu} \leq \mathsf{A}^{\mu} \leq \bar{f}, \tag{11}$$

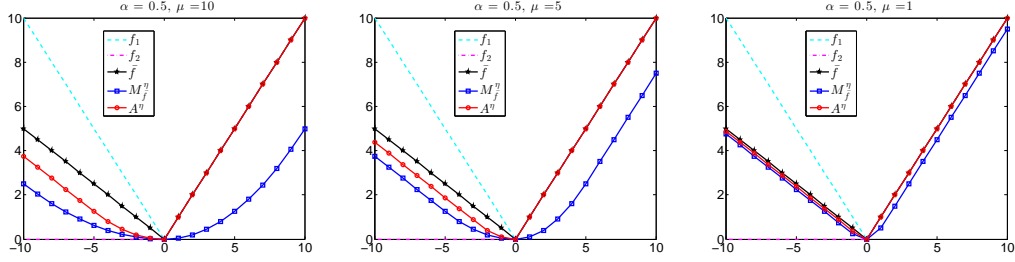

Figure 1: See Example 3 for context. As predicted $\mathsf{M}^\mu_{\mathsf{A}^\mu} \leq \mathsf{A}^\mu \leq \bar{f}$. Observe that the proximal average $\mathsf{A}^\mu$ remains nondifferentiable at 0 while $\mathsf{M}^\mu_{\mathsf{A}^\mu}$ is smooth everywhere. For $x \geq 0$, $f_1 = f_2 = \bar{f} = \mathsf{A}^\mu$ (the red circled line), thus the proximal average $\mathsf{A}^\mu$ is a strictly tighter approximation than smoothing. When $\mu$ is small (right panel), $\bar{f} \approx \mathsf{M}^\mu_{\mathsf{A}^\mu} \approx \mathsf{A}^\mu$.

meaning that the proximal average $\mathsf{A}^\mu$ is a better under-approximation of $\bar{f}$ than $\mathsf{M}^\mu_{\mathsf{A}^\mu}$.

Let us compare the proximal average $\mathsf{A}^\mu$ with the smooth approximation $\mathsf{M}^\mu_{\mathsf{A}^\mu}$ on a 1-D example.

**Example 3.** *Let* $f_1(x) = |x|$, $f_2(x) = \max\{x, 0\}$. *Clearly both are* 1-*Lipschitz continuous. Moreover,* $\mathsf{P}^\mu_{f_1}(x) = \operatorname{sign}(x)(|x| - \mu)_+$, $\mathsf{P}^\mu_{f_2}(x) = (x - \mu)_+ + x - (x)_+$,

$$\mathsf{M}^\mu_{f_1}(x) = \begin{cases} \frac{x^2}{2\mu}, & |x| \leq \mu \\ |x| - \mu/2, & otherwise \end{cases}, \text{ and } \mathsf{M}^\mu_{f_2}(x) = \begin{cases} 0, & x \leq 0 \\ \frac{x^2}{2\mu}, & 0 \leq x \leq \mu \\ x - \mu/2, & otherwise \end{cases}.$$

*Finally, using* (9) *we obtain (with* $\alpha_1 = \alpha, \alpha_2 = 1 - \alpha$*)*

$$\mathsf{A}^\mu(x) = \begin{cases} x, & x \geq 0 \\ \frac{\alpha}{1-\alpha}\frac{x^2}{2\mu}, & (\alpha - 1)\mu \leq x \leq 0 \\ -\alpha x - (1 - \alpha)\frac{\alpha\mu}{2}, & x \leq (\alpha - 1)\mu \end{cases}.$$

*Figure* 1 *depicts the case* $\alpha = 0.5$ *with different values of the smoothing parameter* $\mu$.

# 4 Theoretical Justification

Given our development in the previous section, it is now clear that our proposed algorithm aims at solving the approximation

$$\min_x \ell(x) + \mathsf{A}^\mu(x). \tag{12}$$

The next important piece is to show how a careful choice of $\mu$ would lead to a strictly better convergence rate than S-APG.

Recall that using APG to slove (12) requires computing the following proximal map in each iteration:

$$\mathsf{P}^{1/L_0}_{\mathsf{A}^\mu}(x) = \operatorname*{argmin}_y \frac{L_0}{2}\|x - y\|^2 + \mathsf{A}^\mu(y),$$

which, unfortunately, is not yet amenable to efficient computation, due to the mismatch of the constants $1/L_0$ and $\mu$ (recall that in the decomposition (7) the superscript and subscript must both be $\mu$). In general, there is no known explicit formula that would reduce $\mathsf{P}^{1/L_0}_f$ to $\mathsf{P}^\mu_f$ for different positive constants $L_0$ and $\mu$ [17, p. 338], see also iv) in Proposition 1. Our fix is almost trivial: If necessary, we use a bigger Lipschitz constant $L_0 = 1/\mu$ so that we can compute the proximal map easily. This is indeed legitimate since $L_0$-Lipschitz implies $L$-Lipschitz for any $L \geq L_0$. Said differently, all we need is to tune down the stepsize a little bit in APG. We state formally the convergence property of our algorithm as (Proof in Appendix C):

**Theorem 1.** *Fix the accuracy* $\epsilon > 0$. *Under Assumption* 1 *and the choice* $\mu = \min\{1/L_0, 2\epsilon/\overline{M^2}\}$, *after at most* $\sqrt{\frac{2}{\mu\epsilon}}\|x_0 - x\|$ *steps, the output of Algorithm* 1, *say* $\tilde{x}$, *satisfies*

$$\ell(\tilde{x}) + \bar{f}(\tilde{x}) \leq \ell(x) + \bar{f}(x) + 2\epsilon.$$

*The same guarantee holds for Algorithm* 2 *after at most* $\frac{1}{2\mu\epsilon}\|x_0 - x\|^2$ *steps.*

Note that if we could reduce $\mathsf{P}_{\mathsf{A}^\mu}^{1/L_0}$ efficiently to $\mathsf{P}_{\mathsf{A}^\mu}^\mu$, we would end up with the optimal (overall) rate $O(\sqrt{1/\epsilon})$, even though we approximate the nonsmooth function $\bar{f}$ by the proximal average $\mathsf{A}^\mu$. In other words, approximation itself does not lead to an inferior rate. It is our incapability to (efficiently) relate proximal maps that leads to the sacrifice in convergence rates.

## 5  Discussions

To ease our discussion with related works, let us first point out a fact that is not always explicitly recognized, that is, S-APG essentially relies on approximating the nonsmooth function $\bar{f}$ with $\mathsf{M}_{\mathsf{A}^\mu}^\mu$. Indeed, consider first the case $K = 1$. The smoothing idea introduced in [8] purports the superficial max-structure assumption, that is, $f(x) = \max_{y \in C} \langle x, y \rangle - h(y)$ where $C$ is some bounded convex set and $h \in \Gamma_0$. As it is well-known (also easily verified from definition), $f \in \Gamma_0$ is $M$-Lipschitz continuous (*w.r.t.* the norm $\| \cdot \|$) iff $\mathrm{dom}\, f^* \subseteq \mathsf{B}_{\|\cdot\|}(\mathbf{0}, M)$, the ball centered at the origin with radius $M$. Thus the function $f \in \Gamma_0$ admits the max-structure iff it is Lipschitz continuous, *i.e.*, satisfying our Assumption 1, in which case $h = f^*$ and $C = \mathrm{dom}\, f^*$. [8] proceeded to add some "distance" function $\mathsf{d}$ to obtain the approximation $f_\mu(x) = \max_{y \in C} \langle x, y \rangle - f^*(y) - \mu \mathsf{d}(y)$. For simplicity, we will only consider $\mathsf{d} = \mathsf{q}$, thus $f_\mu = (f^* + \mu \mathsf{q})^* = \mathsf{M}_f^\mu$. The other assumption of S-APG [8] is that $f_\mu$ and the maximizer in its expression can be easily computed, which is precisely our Assumption 2. Finally for the general case where $\bar{f}$ is an average of $K$ nonsmooth functions, the smoothing technique is applied in a component by component way, *i.e.*, approximate $\bar{f}$ with $\mathsf{M}_{\mathsf{A}^\mu}^\mu$.

For comparison, let us recall that S-APG finds a $2\epsilon$ accurate solution in at most $O(\sqrt{L_0 + \overline{M^2}/(2\epsilon)}\sqrt{1/\epsilon})$ steps since the Lipschitz constant of the gradient of $\ell + \mathsf{M}_{\mathsf{A}^\mu}^\mu$ is upper bounded by $L_0 + \overline{M^2}/(2\epsilon)$ (under the choice of $\mu$ in Theorem 1). This is *strictly* worse than the complexity $O(\sqrt{\max\{L_0, \overline{M^2}/(2\epsilon)\}}\sqrt{1/\epsilon})$ of our approach. In other words, we have managed to remove the secondary term in the complexity bound of S-APG. We should emphasize that this *strict* improvement is obtained under exactly the same assumptions and with an algorithm as simple (if not simpler) as S-APG. In some sense it is quite remarkable that the seemingly "naive" approximation that pretends the linearity of the proximal map not only can be justified but also leads to a strictly better result.

Let us further explain how the improvement is possible. As mentioned, S-APG approximates $\bar{f}$ with the smooth function $\mathsf{M}_{\mathsf{A}^\mu}^\mu$. This smooth approximation is beneficial if our capability is limited to smooth functions. Put differently, S-APG implicitly treats applying the fast *gradient* algorithms as the ultimate goal. However, the recent advances on nonsmooth optimization have broadened the range of fast schemes: It is not smoothness but the proximal map that allows fast convergence. Just as how APG improves upon the subgradient method, our approach, with the ultimate goal to enable efficient computation of the proximal map, improves upon S-APG. Another lesson we wish to point out is that unnecessary "over-smoothing", as in S-APG, does hurt the performance since it always increases the Lipschitz constant. To summarize, smoothing is not free and it should be used when truly needed.

Lastly, we note that our algorithm shares some similarity with forward-backward splitting procedures and alternating direction methods [9, 18, 19], although a detailed examination will not be given here. Due to space limits, we refer further extensions and improvements to [20, Chapter 3].

## 6  Experiments

We compare the proposed algorithm with S-APG on two important problems: overlapping group lasso and graph-guided fused lasso. See Example 1 and Example 2 for details about the nonsmooth function $\bar{f}$. We note that S-APG has been demonstrated with superior performance on both problems in [13], therefore we will only concentrate on comparing with it. Bear in mind that the purpose of our experiment is to verify the theoretical improvement as discussed in Section 5. We are not interested in fine tuning parameters here (despite its practical importance), thus for a fair comparison, we use the same desired accuracy $\epsilon$, Lipschitz constant $L_0$ and other parameters for all methods. Since both our method and S-APG have the same per-step complexity, we will simply run them for a maximum number of iterations (after which saturation is observed) and report all the intermediate objective values.

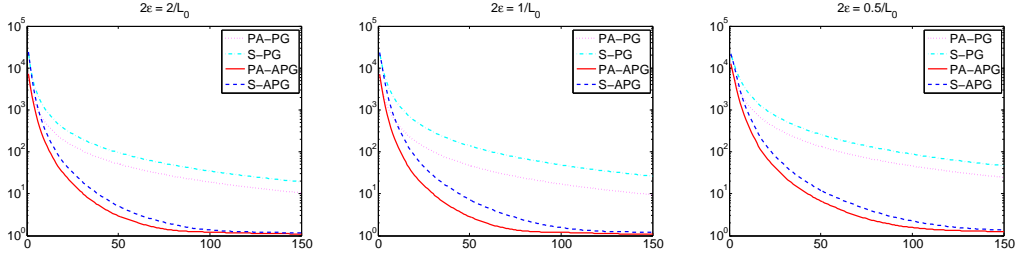

Figure 2: Objective value vs. iteration on overlapping group lasso.

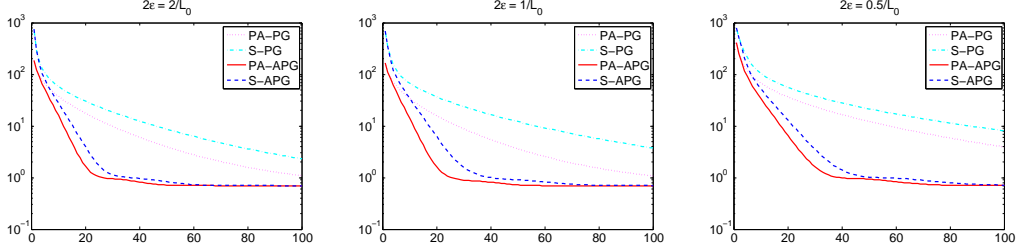

Figure 3: Objective value vs. iteration on graph-guided fused lasso.

**Overlapping Group Lasso:** Following [13] we generate the data as follows: We set $\ell(x) = \frac{1}{2\lambda K}\|Ax - b\|^2$ where $A \in \mathbb{R}^{n \times d}$ whose entries are sampled from *i.i.d.* normal distributions, $x_j = (-1)^j \exp(-(j-1)/100)$, and $b = Ax + \xi$ with the noise $\xi$ sampled from zero mean and unit variance normal distribution. Finally, the groups in the regularizer $\bar{f}$ are defined as

$$\{\{1, \ldots, 100\}, \{91, \ldots, 190\}, \ldots, \{d - 99, \ldots, d\}\},$$

where $d = 90K + 10$. That is, there are $K$ groups, each containing 100 variables, and the groups overlap by 10 consecutive variables. We adopt the uniform weight $\alpha_k = 1/K$ and set $\lambda = K/5$.

Figure 2 shows the results for $n = 5000$ and $K = 50$, with three different accuracy parameters. For completeness, we also include the results for the non-accelerated versions (PA-PG and S-PG). Clearly, accelerated algorithms are much faster than their non-accelerated cousins. Observe that our algorithms (PA-APG and PA-PG) converge consistently faster than S-APG and S-PG, respectively, with a big margin in the favorable case (middle panel). Again we emphasize that this improvement is achieved without any overhead.

**Graph-guided Fused Lasso:** We generate $\ell$ similarly as above. Following [13], the graph edges $E$ are obtained by thresholding the correlation matrix. The case $n = 5000, d = 1000, \lambda = 15$ is shown in Figure 3, under three different desired accuracies. Again, we observe that accelerated algorithms are faster than non-accelerated versions and our algorithms consistently converge faster.

## 7   Conclusions

We have considered the composite minimization problem which consists of a smooth loss and a sum of nonsmooth regularizers. Different from smoothing, we considered a seemingly naive *nonsmooth* approximation which simply pretends the linearity of the proximal map. Based on the proximal average, a new tool from convex analysis, we proved that the new approximation leads to a novel algorithm that *strictly* improves the state-of-the-art. Experiments on both overlapping group lasso and graph-guided fused lasso verified the superiority of the proposed method. An interesting question arose from this work, also under our current investigation, is in what sense certain approximation is optimal? We also plan to apply our algorithm to other practical problems.

## Acknowledgement

The author thanks Bob Williamson and Xinhua Zhang from NICTA—Canberra for their hospitality during the author's visit when this work was performed; Warren Hare and Yves Lucet from UBC—Okanagan for drawing his attention to the proximal average; and the reviewers for their valuable comments.

## Footnotes

[1]In this paper we satisfy ourselves with convergence in terms of function values, although with additional assumptions/efforts it is possible to argue for convergence in terms of the iterates.

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
