[Supplementary Material · pa_appendix.pdf]

# A    Proof of Proposition 2

*Proof.* Fix $f, g \in \Gamma_0$. First note that the Fenchel conjugation enjoys (and is characterized by) the order reversing property:
$$f \geq g \iff f^* \leq g^*.$$

Since $(\mathsf{M}_f^\mu)^* = f^* + \mu\mathsf{q} \in \mathsf{SC}_\mu$ we have $\mathsf{M}_f^\mu \in \mathsf{SS}_{1/\mu}$. On the other hand, let $h \in \mathsf{SS}_{1/\mu}$. Then $g = h^* - \mu\mathsf{q} \in \Gamma_0$, hence $h^* = g + \mu\mathsf{q}$ and $h = (g + \mu\mathsf{q})^* = \mathsf{M}_{g^*}^\mu$. Therefore $\mathsf{M}^\mu$ is onto.

It should be clear that $\mathsf{M}^\mu : \Gamma_0 \to \mathsf{SS}_{1/\mu}$ is increasing *w.r.t.* the pointwise order, *i.e.*, $f \geq g \implies \mathsf{M}_f^\mu \geq \mathsf{M}_g^\mu$. On the other hand, $\mathsf{M}_f^\mu \geq \mathsf{M}_g^\mu \implies (\mathsf{M}_f^\mu)^* \leq (\mathsf{M}_g^\mu)^*$, which, by i) in Proposition 1, means $f^* + \mu\mathsf{q} \leq g^* + \mu\mathsf{q} \implies f^* \leq g^* \implies f = f^{**} \geq g^{**} = g$. Hence $\mathsf{M}^\mu$ is an injection.

Let $\alpha \in ]0, 1[$, then
$$
\begin{aligned}
\mathsf{M}_{\alpha f + (1-\alpha)g}^\mu(x) &= \min_y \tfrac{1}{2\mu}\|x - y\|^2 + \alpha f(y) + (1 - \alpha)g(y) \\
&= \min_y \tfrac{\alpha}{2\mu}\|x - y\|^2 + \alpha f(y) + \tfrac{1-\alpha}{2\mu}\|x - y\|^2 + (1 - \alpha)g(y) \\
&\geq \min_y \tfrac{\alpha}{2\mu}\|x - y\|^2 + \alpha f(y) + \min_y \tfrac{1-\alpha}{2\mu}\|x - y\|^2 + (1 - \alpha)g(y) \\
&= \alpha\mathsf{M}_f^\mu(x) + (1 - \alpha)\mathsf{M}_g^\mu(x),
\end{aligned}
$$
verifying the concavity of $\mathsf{M}^\mu$. $\qquad\qquad\square$

# B    Proof of Proposition 3

*Proof.* First observe that by the definition of the proximal average
$$\bar{f} - \mathsf{M}_{\mathsf{A}^\mu}^\mu = \sum_k \alpha_k(f_k - \mathsf{M}_{f_k}^\mu) \geq 0,$$

since $f \geq \mathsf{M}_f^\mu$ for any $f \in \Gamma_0$. On the other hand
$$
\begin{aligned}
\sup_x f_k(x) - \mathsf{M}_{f_k}^\mu(x) &= \sup_x f_k(x) - \min_y \tfrac{1}{2\mu}\|x - y\|^2 + f_k(y) \\
&= \sup_{x,y} f_k(x) - f_k(y) - \tfrac{1}{2\mu}\|x - y\|^2 \\
&\leq \sup_{x,y} M_k\|x - y\| - \tfrac{1}{2\mu}\|x - y\|^2 \\
&\leq \tfrac{\mu M_k^2}{2},
\end{aligned}
$$
where the first inequality is due to the Lipschitz assumption on $f_k$. Therefore
$$\sup_x \bar{f}(x) - \mathsf{M}_{\mathsf{A}^\mu}^\mu(x) \leq \sum_k \alpha_k \left[\sup_x f_k(x) - \mathsf{M}_{f_k}^\mu(x)\right] \leq \tfrac{\mu\overline{M^2}}{2}.$$
$\qquad\qquad\square$

# C    Proof of Theorem 1

*Proof.* Clearly, under our choice of $\mu$, the gradient of $\ell$ is $1/\mu$-Lipschitz continuous (since $1/\mu \geq L_0$). Therefore after at most $\sqrt{\tfrac{2}{\mu\epsilon}}\|x_0 - x\|$ steps the output of Algorithm 1, say $\tilde{x}$, satisfies [2]
$$\ell(\tilde{x}) + \mathsf{A}^\mu(\tilde{x}) \leq \ell(x) + \mathsf{A}^\mu(x) + \epsilon.$$

Then by Proposition 4
$$
\begin{aligned}
[\ell(\tilde{x}) + \bar{f}(\tilde{x})] - [\ell(x) + \bar{f}(x)] &= [\ell(\tilde{x}) + \mathsf{A}^\mu(\tilde{x})] - [\ell(x) + \mathsf{A}^\mu(x)] \\
&\qquad + [\bar{f}(\tilde{x}) - \mathsf{A}^\mu(\tilde{x})] - [\bar{f}(x) - \mathsf{A}^\mu(x)] \\
&\leq \epsilon + \epsilon + 0 = 2\epsilon.
\end{aligned}
$$

The proof for Algorithm 2 is similar. $\qquad\qquad\square$