[Reviews · NeurIPS 2013]

Submitted by Assigned_Reviewer_3

This paper describes a new proximal-gradient based procedure for the minimization of an objective function, composed by the sum of a convex smooth term and several nonsmooth convex (regularization) terms. The motivation comes from the fact that often, and in significant cases, proximal gradient schemes [e.g. FISTA] are not directly applicable to that scenario, since the proximity operator of the sum of several convex functions is not explicitly computable from the proximity operator of each individual term. The proposed algorithm deals directly with these individual prox operators.
The method is actually a mix of approximation techniques (smoothing) as developed by Nesterov [5] and proximal gradient algorithms. Indeed they propose to replace the whole set of nonsmooth terms, with a unique nonsmooth (close) approximation for which the proximity operator is computable; and this approximation turns out to be the proximal average of the regularization terms --- which has bees studied in detail in [11]. Solutions of the approximate problem are then related to the solutions of the original one and a complexity of $O(1/\varepsilon)$ is shown. The authors claim this procedure to be faster then the one based on the smoothing technique (S-APG algorithm [13]). Indeed, although the asymptotic behavior remains the same, an improvement of the constants is proved. This improvement is finally shown in the experimental section.

I find the proposed method quite interesting and the main idea, of approximate with a nicer nonsmooth term instead of smoothing, valuable and original. A weak aspect is that the nonsmooth terms are required to be (globally) Lipschitz continuous, hence finite everywhere. This means that constraints cannot be treated. The theoretical part is generally sound and precise. Unfortunately I am afraid that the proof of the main theorem is not fully correct. Nevertheless the statement is correct and the problem in the proof can be overcome without much effort as I show below. As regards the experimental section, perhaps other comparisons with more recent approaches (like the two loop-algorithm proposed in [10]) could have been useful. Anyway it is clearly stated that the algorithm is meant to improve S-APG and the section of experiments addresses this comparison.


TYPOS, SMALL CORRECTIONS AND SUGGESTIONS:

-- In the properties listed in Prop. 1, I would add the differentiability of the Moreau envelope and the explicit expression of the gradient $\nabla M_f^\mu = \mu^{-1} (Id - P_f^\mu)$.

-- If $SS_\mu$ is the class of $\mu$-Lipschitz continuous functions, then $f \in SC_\mu \iff f^* \in SS_{1/\mu}$ (not $f^* \in SS_\mu$) and $M^\mu : \Gamma_0 \to SS_{1/\mu}$. Moreover $M^\mu : \Gamma_0 \to SS_{1/\mu}$ is concave only on finite valued functions, not on the whole space $\Gamma_0$, since the convex combination of two functions in $\Gamma_0$ might be even not proper. Thus, Prop. 2 should be accordingly corrected.

-- All the results between Def 1 and Prop 3 are in [11]. This should be clearly stated.


A CRITICAL POINT:

I think that the proof of Theorem 1 is not quite correct and should be pursued with more care. The criticism relies on the fact that generally in prox gradient algorithms, the bound on the precision (after a given number of iterations) depends on the distance between the starting point and the solution set of the problem. This means that the notation $O(\sqrt{\frac 1 {\mu \varepsilon}})$, recalled in the proof, actually hides a further dependence on $\mu$ in the constant --- since it is the approximate function that is being minimized. To overcome this problem,
I suggest the authors to follow the path of [13] (see Proof of Theorem 2 in the appendix). Indeed if one chooses $\mu \leq 1/L_0$ and minimizes the approximation $F_\mu(x) = \gamma \ell(x) + A^\mu(x)$ with FISTA, then for every $x \in H$ it holds
$$
F_\mu(x_t) - F_\mu(x) \leq 2 \frac 1 \mu \frac {\Vert x_0 - x \Vert^2}{t^2}\,.
$$
Note that this inequality is valid for every $x \in H$, not only for the minimizers of $F_\mu$ (just look at the proof of FISTA in [2] or [10]). Then if we now take $x_*$ as a minimizer of the original function $F$ and plug it into the previous inequality we end up with
$$
F(x_t) - F(x_*) \leq \frac{\mu \overline{L^2}}{2} + 2\frac{1}{\mu} \frac {\Vert x_0 - x_* \Vert^2}{t^2}.
$$
This way the constant term $\Vert x_0 - x_* \Vert^2$ does not depend anymore from the approximate problem (hence from $\mu$), but only on the original problem (see also the discussion in section 3.6 in [13]). The statement of the theorem remains unchanged.
Summary: I recommend this article for NIPS conference, since it provides still another type of prox gradient procedure that can contribute to complete the study of this family of algorithms.

Submitted by Assigned_Reviewer_4

Instead of solving proximal step exactly, this paper proposes an averaging technique for (accelerated) gradient methods in the presence of linear combination of regularizers. Specifically, when the regularizer is a linear combination of sub-regularizers, one can solve the proximal step independently for each sub-regularizer, and then take a average. Interestingly, it is shown that this simple approach is strictly better than Nesterov's smoothing technique.

There is an important bug in the proof of Proposition 2. The authors claim that "the dual of a \mu-strongly convex function is \mu-smooth" (lines 210 and 495). However, it should be "the dual of a \mu-strongly convex function is (1/\mu)-smooth" (see (i) and (vii) of [16, Theorem 18.15].). As pointed out by the authors (line 211), this proposition is critical for the whole work, and so it should be well fixed.

Parts of the presentation are not very clear. In particular, Nesterov's smoothing technique is an important baseline in this paper, and thus one of the main conclusions (line 264-268) should be elaborated.

Finally, it would be interesting to compare with the recent linearized ADMM methods [1,2], which have also shown to be suitable for applications studied in this paper (namely, overlapping group and graph-guided fused lasso), and they also have the same convergence rate (i.e., O(1/t)) when there is no stochastic variance.

[1] H. Ouyang, N. He, L. Tran, and A. Gray. Stochastic alternating direction method of multipliers. In Proceedings of the 30th International Conference on Machine Learning, Atlanta, GA, USA, 2013.

[2] T. Suzuki. Dual averaging and proximal gradient descent for online alternating direction multiplier method. In Proceedings of the 30th International Conference on Machine Learning, Atlanta, GA, USA, 2013.
Summary: This paper proposed an interesting averaging technique for the proximal methods. However, there is an important bug in the proof of Proposition 2 and parts of the presentation are not very clear.

Submitted by Assigned_Reviewer_5

This paper deals with the problem of minimizing composite objective functions with a smooth loss function and a summation of non-smooth functions. Instead of taking a smooth approximation, the paper suggests a new non-smooth approximation using the idea of proximal averaging, which leads to a strictly better convergence than the smooth counterpart.

Quality and clarity:
It was greatly pleasant to read this paper. The motivation, related works, and contributions are stated clearly and organized very well. Enough technical details were provided to understand the development in analysis. Simple experiments were provided to verify the claimed results.

Originality:
As far as I know, the non-smooth approximation using the proximal averaging is new.

Significant:
I believe this paper brings a very interesting point to the community in optimization perspective, with a new message that a closer approximation is possible than smooth approximations using non-smooth approximations based on proximal averaging, which is easy to compute for many cases.



Summary: This paper suggests a new optimization method using a non-smooth approximation based on proximal averaging, which allows for better approximations to the original non-smooth function than the popular smooth approximation approaches. Thanks to the proximal averaging, this better approximation is achieved without adding any extra computation.
Author Feedback

Author rebuttal: We thank all reviewers for their valuable comments, and we address some of the concerns below.


=================================================
Assigned_Reviewer_3:

Q1: SS_\mu should be SS_{1/\mu}

A: Correct. Will fix this typo. Indeed, we only need to change the definition mu-Lipschitz in line 209 to 1/mu-Lipschitz.


Q2: \Gamma_0 itself is not convex

A: Very true. This pathological case can be eliminated by stating "M is concave on convex subsets of \Gamma_0".


Q3: Results between Def 1 and Prop 3 are not new

A: Correct, although our derivation is slightly different from [11]: [11] is mostly constructive while our intention in this paragraph is to introduce the proximal average to the NIPS audience using a more abstract (hence cleaner) argument. The derived results per se are not new. Will make this more explicit.


Q4: implicit dependence on mu in Theorem 1

A: Very good point, will revise according to the suggestion.


Other comments will be taken into account in the revision.



=================================================
Assigned_Reviewer_4:

Q1: SS_\mu should be SS_{1/\mu}

A: Correct. Will fix this typo. Indeed, we only need to change the definition mu-Lipschitz in line 209 to 1/mu-Lipschitz. We want to emphasize that this is NOT an important bug. Nothing in the proof nor any of our conclusions needs to be changed. It would be a great pity if the reviewer misjudges the paper solely based on this unfortunate typo.

Prop 2 is critical in our more abstract way of defining the proximal average. It is NOT critical for the main results in this paper, since the proximal average has been defined constructively in reference [11]. The reviewer can be assured that the main results in this paper are correct.


Q2: Line 264-268 needs to be elaborated

A: Indeed, we thoroughly explained our conclusion about Nesterov's smoothing idea in section 5, please see line 331-343. Will make the connection more explicit.


Q3: relation with ADMM

A: At the time of submission, we were not aware of the references pointed out by the reviewer, although we did notice the similarity between our approach and ADMM (see line 364-367). We would also like to point out that the main point of this paper is not just another competing algorithm. It is the new way of thinking about approximations that we think are interesting. For instance, we could apply the proposed approximation to the online setting and stochastic setting, and recover similar results obtained in the references (these results were omitted at the time of submission due to the lack of space and the concern not to make the submission unnecessarily long and complicated). It is certainly of interest to study the exact relation between our approach and ADMM, but such comparison (not just experimentally) would require nontrivial further work and we prefer to discuss it later.

We should also point out that the additional references derived their results based on very different techniques than this submission. We will thoroughly study these references and put appropriate citations.



=================================================
Assigned_Reviewer_5:

We thank the reviewer for the positive and precise assessment.